# Prognostic Relevance of RIP140 and ERβ Expression in Unifocal Versus Multifocal Breast Cancers: A Preliminary Report

**DOI:** 10.3390/ijms20020418

**Published:** 2019-01-18

**Authors:** Katharina Müller, Sophie Sixou, Christina Kuhn, Stephan Jalaguier, Doris Mayr, Nina Ditsch, Tobias Weissenbacher, Nadia Harbeck, Sven Mahner, Vincent Cavaillès, Udo Jeschke

**Affiliations:** 1Department of Obstetrics and Gynecology, LMU Munich, University Hospital, 81377 Munich, Germany; kontakt@katharinamueller.net (K.M.); sophie.sixou@univ-tlse3.fr (S.S.); Christina.kuhn@med.uni-muenchen.de (C.K.); nina.ditsch@med.uni-muenchen.de (N.D.); tobias.weissenbacher@med.uni-muenchen.de (T.W.); nadia.harbeck@med.uni-muenchen.de (N.H.); sven.mahner@med.uni-muenchen.de (S.M.); 2Faculté des Sciences Pharmaceutiques, Université Paul Sabatier Toulouse III, 31062 Toulouse Cedex 09, France; 3IRCM, Institut de Recherche en Cancérologie de Montpellier, 34298 Montpellier, France; stephan.jalaguier@inserm.fr (S.J.); vincent.cavailles@inserm.fr (V.C.); 4Department of Pathology, LMU Munich, 80337 Munich, Germany; doris.mayr@med.uni-muenchen.de

**Keywords:** breast cancer, tumor focality, RIP140, LCoR, estrogen receptors

## Abstract

The aim of this study was to investigate the expression of two nuclear receptor transcriptional coregulators, namely RIP140 (receptor-interacting protein of 140 kDa) and LCoR (ligand-dependent corepressor) in unifocal versus multifocal breast cancers. The expression of these two proteins was analyzed by immunohistochemistry in a matched-pair cohort of 21 unifocal and 21 multifocal breast tumors. The expression of the two estrogen receptors (ERα and ERβ) was studied in parallel. RIP140 and LCoR levels appeared lower in unifocal tumors compared to multifocal samples (decreased of immune-reactive scores and reduced number of high expressing cells). In both tumor types, RIP140 and LCoR expression was correlated with each other and with expression of ERβ. Very interestingly, the expression of RIP140, LCoR, and ERβ was inversely correlated with overall survival only for the unifocal group. The negative correlation with overall and recurrence free survival was more pronounced in patients whose unifocal tumors expressed high levels of both RIP140 and ERβ. Altogether, this preliminary report indicates that the ERβ/RIP140 signaling is altered in unifocal breast cancers and correlated with patient outcome. Further investigation is needed to decipher the molecular mechanisms and the biological relevance of this deregulation.

## 1. Introduction

The last decade provided clear evidence for the importance of focality regarding breast cancer (BC) aggressiveness [1,2]. Although a standard classification of focality status is still debated, multifocality was initially defined as two or more separate invasive tumors in the same quadrant and multicentricity as two or more separate invasive tumors in more than one quadrant of the same breast [3]. Nonetheless, since the quadrant definition is not linked to breast anatomy, the difference between multifocality and multicentricity is becoming less and less important. More recently, multifocality has been defined as multiple simultaneous ipsilateral BC lesions, provided they are macroscopically distinct and measurable, irrespective of the localization of the lesions [4]. Multifocality has been associated with lymph node and distant metastases [5,6], with shorter survival [7], and higher mortality rates [8]. Our previous work demonstrated that multicentricity/multifocality was a significant independent predictor for local relapse, distant metastasis, and reduced overall survival (OS) [9]. This previous study emphasized the importance of the combined tumor volume rather than that of the larger tumor diameter. More recently, using a well-balanced matched-pair cohort, we demonstrated a significant lower expression of E-cadherin in the multifocal group [10].

Together with the tumor node metastasis (TNM) classification, the hormone receptor (estrogen receptor ER, progesterone receptor PR) and the HER2 receptor status are standard diagnostic parameters to describe tumor biology and support therapy decisions. For multifocal tumors, assessment of ER, PR, and HER2 status only in the largest lesion (if the lesions do not differ in grade or histological subtype) is still questionable, and an accurate biological characterization of all lesions should be recommended [4]. Regarding estrogen signaling, the expression ratio between ERα and ERβ has a great effect on tumorigenesis in breast and ovarian cancers; ERβ was shown to have inhibiting effects on ERα and thus on cell proliferation. ERβ, especially the isoform 2, is expressed at a higher level than ERα in the normal human mammary gland, but its expression decreased in BC cells, particularly in ERα expressing cells [11,12].

BC signaling and progression is also influenced by the complex interplay of nuclear receptors and their transcriptional coactivators and corepressors [5]. The importance of nuclear receptors and coregulators networks which appear disrupted in BC and have prognostic significance was previously demonstrated [13]. RIP140 (receptor-interacting protein of 140 kDa) is one of the first transcriptional coregulators shown to interact with ERs [14] and to regulate BC cell proliferation and invasion in vitro [15,16,17]. RIP140 was found to be an important transcriptional cofactor for estrogen signaling in ovarian and BC cells [18,19]. The effect of RIP140 appeared stronger on ERβ than on ERα (better in vitro interaction and greater modulation of estradiol-dependent transactivation) [20,21]. Moreover, RIP140 was shown to be more efficiently upregulated by estradiol in ERβ expressing BC cells [22]. Finally, analysis of knock-out mice revealed that RIP140 is an essential factor for normal mammary gland development through its functions on estrogen signaling [23].

LCoR (ligand-dependent corepressor) is another transcriptional coregulator which interacts with agonist-activated ERα and represses its activity via histone deacetylase-dependent and independent mechanisms [24]. The physiological role of LCoR is poorly understood but biological effects on prostate cancer and liver homeostasis have been reported [25,26]. Previously, we reported that LCoR was engaged in a complex with RIP140 and negatively regulated BC cell proliferation in a RIP140-dependent manner [27]. Moreover, we recently analyzed RIP140 and LCoR expression at the protein level in BC biopsies showing that the two proteins were highly correlated in more than 80% of tumors and that RIP140 expression was significantly correlated with patient survival [28].

The present work analyzes the expression of RIP140 and LCoR in BC samples according to their focality status. We used a previously characterized, small but well-balanced matched-pair cohort of 42 unifocal and multifocal BC samples [10]. This cohort is homogenous for the two types of tumors in terms of tumor size, histology grade, lymph node status, and patient survival. Using immunohistochemistry, RIP140 and LCoR expression was monitored in parallel to that of ERα and ERβ, and data were correlated with OS. The results highlight a highly significant negative prognostic impact of RIP140/ERβ coexpression, only in unifocal tumors.

## 2. Results

### 2.1. RIP140 and LCoR Expression in Unifocal vs. Multifocal Tumors

Using immunohistochemistry, we evaluated RIP140 and LCoR expression on 21 matched pairs of unifocal or multifocal tumors described in Table 1.

Semi-quantitative immunoreactive scores (IRS) were quantified by assessing the percentage of positively stained cells together with staining intensity. Figure 1 illustrates RIP140 and LCoR staining results with low or high IRS, in unifocal and multifocal samples. The mean IRS of RIP140 and LCoR expression were compared in unifocal vs. multifocal tumors and found to be lower in the unifocal group compared to the multifocal one, with a significant difference only for LCoR.

As shown in Table 2, the mean IRS for RIP140 was 2.61 in the unifocal group vs. 2.98 in the multifocal, and for LCoR expression, 2.38 vs. 3.38, respectively (*p* < 0.05). Considering IRS values ≥3.25 as positive, the percentage of samples expressing low or high levels of each coregulator was analyzed in the two types of tumors, as shown in Table 2. The majority of the 42 tumors expressed low levels of RIP140, and the percentage of RIP140 high expressing tumors decreased in the unifocal group (23.8% high expressing vs. 42.9% in the multifocal group). Similarly, the percentage of LCoR high expressing tumors decreased in the unifocal group (19% high expressing vs. 57.1% in the multifocal group, *p* = 0.011). Finally, a positive and significant correlation between RIP140 and LCoR expression was observed in the two tumor types (*p* < 0.01) confirming our previous observation [28].

### 2.2. Correlation with Clinical and Biological Parameters

RIP140 and LCoR expression was then correlated with clinical parameters linked to tumor aggressiveness and prognosis. These parameters were the recurrence status, pT, pN, pM, grade, histology types (classified as ductal, lobular, ductal-lobular, medullary, micro papillary, as described in Table 1), as well as expression of the two ER (ERα and ERβ), and HER2. No correlation was observed between RIP140 or LCoR, and most of the clinical and biological parameters analyzed, except for ERβ, as shown in Table 3. Indeed, we observed positive and significant correlations between ERβ and RIP140 or LCoR, both in unifocal and multifocal tumors. The correlations did not appear statistically different in unifocal cases (*r* = 0.741 and 0.783, for RIP140 and LCoR, respectively) and multifocal tumors (*r* = 0.699 and 0.612, respectively).

### 2.3. Correlation with Patient Survival

To analyze whether patient survival was linked to expression of the transcription factors, we then performed Kaplan–Meier analyses using the optimal threshold IRS values determined by receiver operating characteristic curve (ROC-curve) analysis for each selected parameter (as detailed in the “Survival Analysis” section of “Materials and Methods”). In the matched-pair cohort, the mean survival time of patients with unifocal cancer was not significantly different from that of patients with multifocal cancer, as shown in Table 1. As shown in Figure 2, we observed a significant correlation of either high RIP140 (*p* = 0.003), high LCoR (*p* = 0.028), or high ERβ (*p* = 0.001) expression with poor OS in the unifocal group, whereas OS analysis in the multifocal group showed no significant correlation to transcription factor expression levels. Mean survival times in the unifocal group were 153.9 and 53.4 months for low vs. high RIP140 expression, 146.3 and 56.8 months for low vs. high LCoR expression, and 161.9 and 55.7 months for low vs. high ERβ expression, whereas no correlation in either group was observed for ERα expression.

We then combined the parameters to evaluate whether the positivity of two markers could better predict OS. The combination of ERα expression with either RIP140, LCoR, or ERβ expression did not reveal any change in correlation with OS, neither in the unifocal nor in the multifocal group. As illustrated in Figure 3, high LCoR expression combined with high RIP140 expression (A,B) or ERβ expression (C,D) maintained a significant correlation with a poor OS in the unifocal cases only (*p* < 0.01) but did not improve the prognostic impact compared to the individual markers, as shown in Figure 2. In contrast, combined RIP140 and ERβ expression (E,F) had an even stronger correlation with poor OS (*p* = 0.0004), again only in the unifocal group (mean survival times of 160.9 and 44.2 months for low vs. high combined expression).

Taking into account this strong correlation of the combined RIP140 and ERβ expression with poor OS, we calculated the correlation with recurrence free survival (RFS), as shown in Appendix A. We observed again a significant correlation of either high RIP140 (*p* = 0.004) or high ERβ (*p* = 0.0002) expression with poor RFS in the unifocal group (A and C, respectively), whereas RFS analysis in the multifocal group showed no significant correlation (B and D, respectively). The combined RIP140 and ERβ expression (E,F) had again an even stronger correlation with poor RFS (*p* = 0.00001) only in the unifocal group.

## 3. Discussion

The aim of this study was to compare RIP140 and LCoR expression in unifocal and multifocal BC samples, and to identify correlations with clinical parameters. We did not analyze multifocal and multicentric cancers separately but grouped them in the multifocal cohort and compared them to unifocal tumors.

In our matched-pair cohort of BC samples, LCoR appeared expressed at lower levels in the unifocal tumors (lower average IRS values and lower percentage of high expressing tumors). The same trend was noticed for RIP140 although the difference was not significant. Nonetheless, expression of the two transcription factors was highly correlated in both groups, as recently described in a large BC patient cohort [28].

RIP140 or LCoR expression did not correlate with clinicopathological parameters, neither in unifocal nor in multifocal samples, except with ERβ levels. Indeed, a strong and significant correlation with ERβ expression was observed for both coregulators in the unifocal and in the multifocal group. In contrast, no significant correlations were observed between RIP140, LCoR, or ERβ expression and cell adhesion-related glycoproteins, namely E-cadherin, MUC1, and β-catenin compared in the unifocal vs. multifocal groups (data not shown). We previously quantified these three proteins in the same matched cohort, demonstrating that cytoplasmic β-catenin was associated significantly with reduced OS in unifocal patients [10].

Survival analysis demonstrated that high LCoR expression, and to a stronger extent high RIP140 or ERβ expression, correlated significantly with poor OS only in the unifocal BC samples. No significant correlation with survival was seen in the multifocal BC cohort, indicating a difference in the expression patterns and prognostic relevance of these transcription factors according to the focality status. It should be mentioned again that other biological parameters, such as tumor size, histology grade, and lymph node status, were similar between the two groups.

Concerning RIP140, these results are in accordance with our data obtained on a cohort of 320 samples showing that high RIP140 protein levels were correlated with short disease-free survival (DFS) [28]. With respect to ERβ, the biological relevance of high expression and its consequences on clinical outcome are still controversially discussed, depending on patient cohorts, analysis method (ERβ mRNA or protein) or ERβ isoforms tested [29,30,31,32]. We analyzed ERβ1, the full-length fully functional ERβ isoform [29]. Indeed, ERβ overexpression was found to be correlated either with poor [33,34] or with favorable [35] DFS. A study performed on 139 ER-positive BC samples demonstrated that ERβ protein levels were correlated with small tumor size, while ERβ mRNA levels were associated with poor DFS and were found to be an independent predictor of disease recurrence [33]. However, none of these studies analyzed survival according to tumor focality.

Interestingly, our data demonstrated an enhanced prognostic impact of combined ERβ/RIP140 expression on OS. RIP140 exerts a strong inhibitory effect on estrogen signaling; it was previously shown to interact more efficiently with ERβ than with ERα and to inhibit its activity with greater efficacy [20]. Cistrome and transcriptome analyses combined with clustering algorithms also supported a preferential recruitment of RIP140 by ERβ [22]. Moreover, the induction of RIP140 appeared mainly driven by ERβ in ovarian cancer cells [20]. Being aware of the small number of patients in our study, and the limitations of the immunostaining as a semi-quantitative assessment technique, we looked for confirmation of this significant effect on unifocal BC patients. We could confirm the negative impact of RIP140 or ERβ alone, and of the combined high ERβ/RIP140 expression on DFS also, suggesting that the two proteins may control various aspects of BC progression.

## 4. Materials and Methods

### 4.1. Collective

As previously described [10], our total collective was formed of a consecutive patient cohort consisting of 112 patients documented and surgically treated for primary BC between 2000 and 2002 at the Department of Gynecology of the University Hospital in Munich-Innenstadt; 57 unifocal BC patients and 55 patients with multicentric/multifocal disease. Data were entered into the database in an anonymized and coded fashion.

Because of the uneven distribution of prognostic factors in our original patient group, a matched pair analysis was performed. From this collective, two equivalent groups of 23 BC patients with multicentric/multifocal vs. unifocal tumors were selected according to the highest degree of equivalence in the following hierarchical and sequential order: tumor size at the time of primary diagnosis, histology grading, and lymph node status. We deliberately matched patients based on the criteria at the time of primary diagnosis. Kruskal–Wallis one-way analysis of variance was used, which tests the equality of population medians among groups in a non-parametric way (continuation of the Mann–Whitney U test to analyze ≥3 groups). Hereby BC were equally allocated into the two groups (*p* = 1.000). The Institutional Review Board of the Ludwig Maximilian University (LMU) Munich, Germany, approved the study (approval number 048–08, 18 03. 2008) and all the patients gave informed consent. For the present study, from these 2 groups of 23 patients, only tumors of 2 groups of 21 patients could be stained for RIP140. Our study was then based upon a total of 42 primary BC, with 21 tumors classified as primary unifocal BC and 21 as multifocal or multicentric BC.

The focality status was evaluated by clinical examination, ultrasound, and mammography. In some cases, further investigation including nuclear magnetic resonance imaging (NMRI), galactography, or pneumocystography was necessary to accurately describe the focality status. Tumors with unconfirmed or questionable focality status (prior to histological examination) were excluded. All patients included in this project had to be free of any other disease at the time of the primary diagnosis and to be treated for resectable BC.

Tumor stage at primary diagnosis was histologically evaluated using the “Union internationale contre le cancer” (UICC) TNM classification which includes tumor size (primary tumor size, or pT, classified as: pT1a-c, pT2, pT3, pT4a-d), involvement of regional lymph nodes (N), and presence or absence of metastases (M). The tumor grade was determined by an experienced pathologist (Dr D. Mayr) of the LMU Department of Pathology and classified according to the WHO (Nottingham grading respectively to Elston and Ellis modification of Bloom–Richardson grading) [36]. For multifocal samples, each tumor was analyzed and classified as ERα or HER2 positive BC if at least one tumor lesion was positive. Additional data, such as age, ERα status, HER2-status, histological grade, metastases, local recurrence, progression, and survival, were retrieved from the Munich Cancer Registry.

All patient data were fully anonymized, and all diagnostic procedures had already been fully completed when samples were collected for the study. Authors were blinded from the clinical information during the experimental analysis. This study was approved by the Ethical Committee of the Medical Faculty, LMU, Munich, Germany and informed consent was obtained from all patients.

### 4.2. Immunohistochemistry

Expression of ERα and HER2 was determined at diagnosis, in all BC samples of this cohort at the LMU Department of Pathology, Germany. ERα expression was evaluated by immunohistochemistry, as previously described [36]. Samples showing nuclear staining in more than 10% of tumor cells were considered as hormone receptor-positive, in agreement with the guidelines at the time of the analysis (2000–2002). HER2 expression was analyzed with an automated staining system (Ventana; Roche, Mannheim, Germany), according to the manufacturer’s instructions. Data on MUC-1, β-catenin, and E-cadherin expression in these BC samples were extracted from a previously published study [10]. For ERβ, RIP140, and LCoR analysis, samples were processed as previously described [28,37,38]. Formalin-fixed and paraffin-embedded sections of 3 µm were deparaffinized using xylol for 20 minutes, rehydrated in a descending ethanol gradient (100%, 96%, and 70%) and subjected to epitope retrieval for 5 min in a pressure cooker using sodium citrate buffer (pH 6.0). After returning to room temperature, sections were washed twice in PBS (phosphate buffered saline). The sections were immersed in 3% H_2_O_2_ in methanol for 20 min to block endogenous peroxidase activity. To prevent non-specific binding of the primary antibody, the sections were treated by the appropriate blocking solution. Incubation with the primary antibody (RIP140: polyclonal antibody, Sigma Aldrich; LCoR: polyclonal antibody, Novus Biologicals; ERβ1: monoclonal antibody, Dako, Glostrup) was performed for 16 hours at a temperature of 8 °C. After washing with PBS, reactivity was detected by the Vectastain Elite ABC-Kit (Vector Laboratories, Burlingame, CA, USA) according to the producer’s protocol. Visualization was reached with DAB substrate and chromogen (3, 3´-diaminobenzidine DAB, Dako, Glostrup, Denmark) for 2 minutes. Then the slides were counterstained with Maier’s acidic hematoxylin and dehydrated in an ascending alcohol series (50–98%), then immersed in xylol. The sections were embedded and covered. Placenta tissue served as positive control staining. Replacement of the primary antibody with mouse or rabbit IgG was used as negative control. 

The slides were investigated using a Leitz Diaplan microscope (Wetzlar, Germany) with a 3CCD color camera (JVC, Victor Company of Japan, Yokohama, Japan). To differentiate the intensity and distribution patterns, the semi-quantitative IRS was used. The IRS assesses the percentage of positively stained cells (graded as 0 = none, 1 = weak, 2 = moderate, and 3 = strong) with the cells´ intensity of staining (0 = no staining, 1 = <10% of cells, 2 = 11–50% of cells, 3 = 51–80%, and 4 = >81% of cells) by multiplying. The reproducibility of RIP140 and LCoR stainings was checked by triplicate stainings of some sections and for all sections, the stainings were analyzed by two independent observers.

### 4.3. Statistical Analyses

Statistical analyses were performed on the 21 matched pairs resulting in a total collective of 42 patients. Besides the collective characterization described above, the differences were calculated using mean or percentage bilateral analysis, as shown in Table 2, and the correlations using Spearman’s Rho test, as shown in Table 2 and Table 3. All differences or correlations are statistically significant for *p* < 0.05 (*), *p* < 0.01 (**) or *p* < 0.001 (***). The IBM Statistical Package for the Social Sciences 24.0 (SPSS Inc., Chicago, IL, USA) was used to test for statistical significance.

### 4.4. Survival Analysis

To compare the mean immunoreactivity levels described by the IRS, the groups were divided into low vs. high expressing (RIP140, LCoR, ERβ, and ERα) cases. Therefore ROC-curve analyses were performed and the maximum difference between sensitivity and specificity was used for identification of the cut-off level for RIP140, LCoR, ERβ, and ERα. The following thresholds were determined regarding OS: RIP140 ≥ 3.25, LCoR ≥ 3.25, ERβ ≥ 5.25, ERα ≥ 3.0. These thresholds were used to determine the percentages of low or high RIP140 and LCoR expression described in Table 2, besides the survival analysis.

Survival times were compared by Kaplan–Meier graphics and differences in OS, as shown in Figure 2 and Figure 3, and DFS, as shown in Appendix A, were tested for significance using the chi-square statistics of the log rank test. Data were assumed to be statistically significant in case of *p*-value < 0.05. Kaplan–Meier graphics were then provided for each subgroup and each marker as well as for the combined expression of two markers in order to compare the differences of survival times between unifocal and multifocal tumors and between low and high receptor expression. Only mean survival times are presented, as median survival times were not reached for all sub-groups.

## 5. Conclusions

In conclusion, this preliminary report shows that the prognostic value of ERβ/RIP140 coexpression differs according to tumor focality and significantly correlates with poor OS and DFS only in patients with unifocal BC. While being small, our cohort presents the great advantage of being a matched-pair cohort with perfect matching criteria. Moreover, despite the relatively limited number of cases, the results obtained in unifocal tumors for RIP140 and ERβ were highly significant. These data strengthen the need to further investigate the relevance of these two genes as independent prognostic markers in extended cohorts and to enlarge the analysis to other nuclear receptors and coregulators. It would also be of interest to understand why ERβ and RIP140 lose their prognostic impact (as single markers or in combination) in multifocal BC. Altogether, our results may lead to a better understanding of key transcription networks involved in multifocal BC and to define the clinical potential of new biological markers.

## Figures and Tables

**Figure 1 ijms-20-00418-f001:**
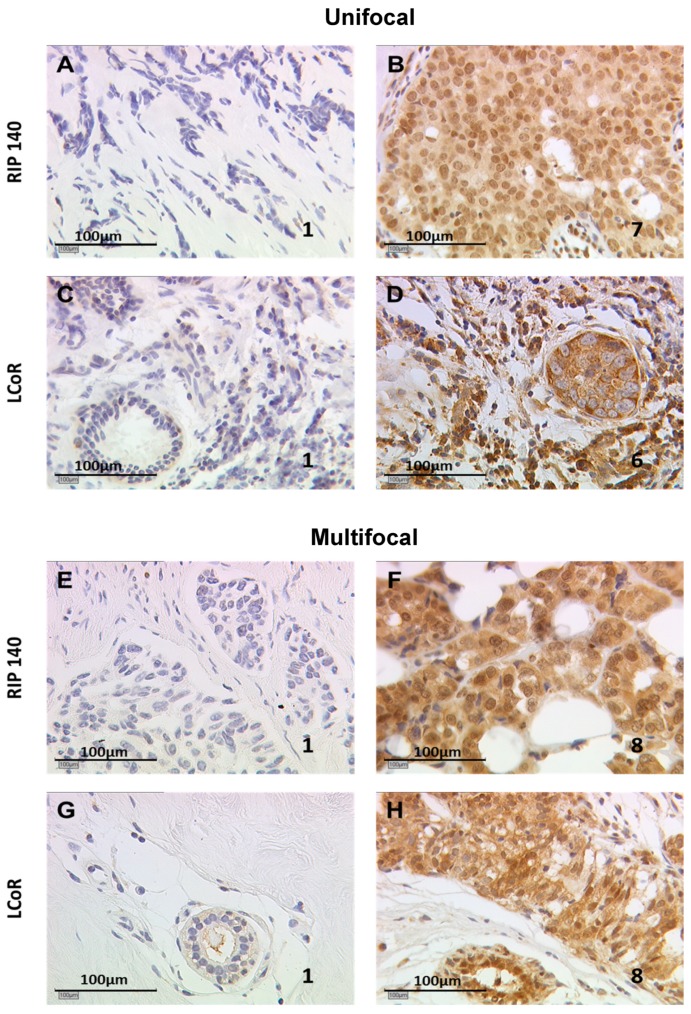
Immunohistological staining of receptor interacting protein of 140 kDa (RIP140) and ligand dependent corepressor (LCoR) in unifocal or multifocal breast cancers. RIP140 (**A**,**B**,**E**,**F**), and LCoR (**C**,**D**,**G**,**H**) expression was evaluated by immunohistochemistry in unifocal (**A**–**D**) and multifocal (**E**–**H**) cases. Positive staining appears in brown color, nuclei appear in blue. Examples of low expression (**A**,**C**,**E**,**G**) vs. high expression (**B**,**D**,**F**,**H**) have been selected (immunoreactive score (IRS) values indicated in each photograph).

**Figure 2 ijms-20-00418-f002:**
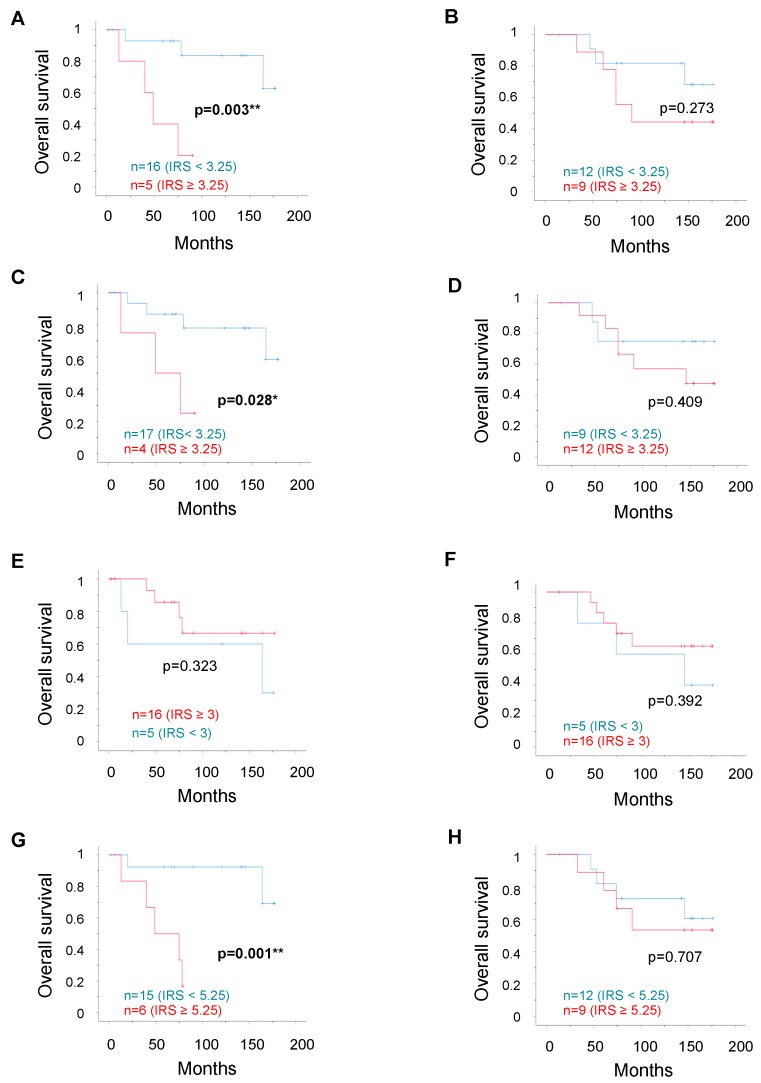
Patient overall survival (OS) according to RIP140, LCoR, ERα, or ERβ expression. Kaplan–Meier analysis of the correlation between RIP140 (**A**,**B**), LCoR (**C**,**D**), ERα (**E**,**F**), or ERβ (**G**,**H**) expression with OS in the matched pairs of unifocal (**A**,**C**,**E**,**G**) or multifocal (**B**,**D**,**F**,**H**) breast cancers. The IRS cut-off values together with the number of cases in each arm are indicated in each panel. Correlations are statistically significant for *p* < 0.05 (*) and for *p* < 0.01 (**).

**Figure 3 ijms-20-00418-f003:**
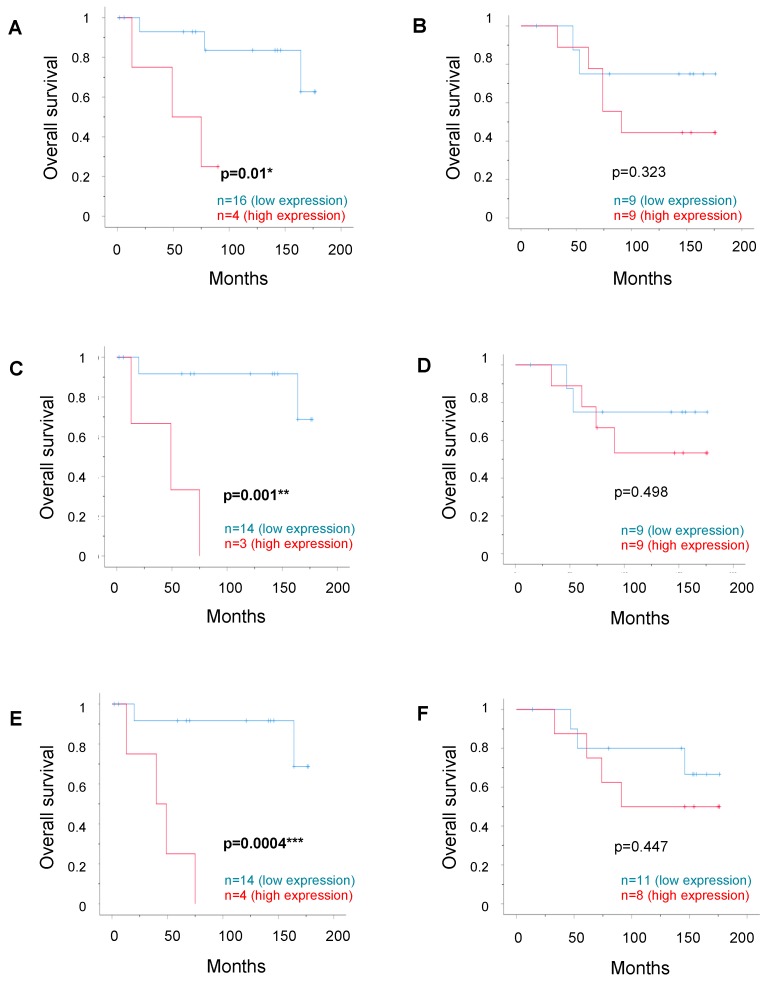
Overall survival according to combined expression of RIP140, LCoR, or ERβ. Kaplan–Meier analysis of the correlation between combined low vs. high expression of two markers with OS in the matched pairs (unifocal for **A**, **C**, **E** or multifocal for **B**, **D**, **F**). The analysis was performed with LCoR expression combined with RIP140 (**A**,**B**) or ERβ (**C**,**D**) expression, or with RIP140 expression combined with ERβ expression (E, F). Correlations are statistically significant for *p* < 0.05 (*), for *p* < 0.01 (**), and for *p* < 0.001 (***).

**Table 1 ijms-20-00418-t001:** Clinicopathological features of the matched-pair cohort.

Parameters	Unifocal (*n* = 21)	Multifocal (*n* = 21)
**Age**		
Mean (years)	58.5	63.7
**Histological type**		
Ductal	13 (61.9%)	16 (71.4%)
Lobular	3 (14.3%)	4 (19%)
Ductal-Lobular	1 (4.8%)	2 (9.5%)
Medullary	1 (4.8%)	0 (0%)
Micropapillary	2 (9.5%)	0 (0%)
Unknown	1 (4.8%)	0 (0%)
**Tumor size**		
pT1 a, b, c	18 (85.7%)	18 (85.7%)
pT2	2 (9.5%)	2 (9.5%)
pT3	0 (0%)	0 (0%)
pT4a, b, c, d	1 (4.8%)	1 (4.8%)
**Grade**		
I	0 (0%)	0 (0%)
II	17 (81.0%)	17 (81.0%)
III	4 (19.0%)	4 (19.0%)
**Lymph node metastasis**		
No	16 (76.2%)	16 (76.2%)
Yes	4 (19%)	4 (19%)
Unknown	1 (4.8%)	1 (4.8%)
**Local recurrence**		
No	13 (61.9%)	17 (81%)
Yes	8 (38.1%)	4 (19%)
**Overall survival**		
Mean time (months)	130.79	133.41
**ERα status**		
Negative	5 (23.8%)	5 (23.8%)
Positive	16 (76.2%)	16 (76.2%)
**ERβ status**		
Negative	10 (47.6%)	5 (23.8%)
Positive	11 (52.4%)	16 (76.2%)
**HER2 status**		
Negative	18 (85.7%)	16 (76.2%)
Positive	3 (14.3%)	5 (23.8%)

**Table 2 ijms-20-00418-t002:** Distribution and correlation of RIP140 and LCoR expression in unifocal versus multifocal breast cancers.

Parameters	Unifocal (*n* = 21)	Multifocal (*n* = 21)
**RIP140 expression**		
Mean IRS ± SE	2.61 ± 1.74	2.98 +/− 2.12
Low expressing tumors n (%)	16 (76.2%)	12 (57.1%)
High expressing tumors n (%)	5 (23.8%)	9 (42.9%)
**LCoR expression**		
Mean IRS ± SE	2.38 * ± 1.49	3.38 ± 1.93
Low expressing tumors n (%)	17 * (80.1%)	9 (42.9%)
High expressing tumors n (%)	4 * (19%)	12 (57.1%)
**Correlation between RIP140 and LCoR**		
Spearman’s Rho correlation coefficient	0.714 **	0.686 **

The cut-off value between low and high expression is defined for RIP140 and LCoR as an IRS ≥ 3.25. The difference or correlation are statistically significant for *p* < 0.05 (*) and for *p* < 0.01 (**), using mean or percentage bilateral analysis and Spearman’s Rho test.

**Table 3 ijms-20-00418-t003:** Correlation analysis of RIP140 and LCoR expression with clinical parameters and estrogen receptor (ER) expression in unifocal versus multifocal breast cancers.

	RIP140		LCoR	
	Unifocal	Multifocal	Unifocal	Multifocal
Recurrence Status	0.368	−0.01	0.353	0.091
pT	−0.157	0.083	0.017	0.091
pN	−0.204	−0.054	0.018	−0.096
pM	−0.257	0.159	−0.34	0.011
Grade	−0.152	−0.01	0.071	0.081
Histology	0.034	0.124	−0.12	0.004
ERα	0.012	0.126	0.181	−0.212
ERβ	0.741 **	0.699 **	0.783 **	0.612 **
HER2	0.397	0.196	0.239	0.309

Spearman’s Rho correlation coefficient are presented. Correlation are statistically significant for *p* < 0.01 (**), using a Spearman’s Rho test. pT: primary tumor size; pN: lymph node involvement; pM: state of metastasis.

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
