# Peer review of "Prognostic Relevance of RIP140 and ERβ Expression in Unifocal Versus Multifocal Breast Cancers: A Preliminary Report"

_ijms, 2019, doi:10.3390/ijms20020418_

Reviewer 1 Report

The authors analyzed the RIP140 and LCoR levels by immunohistochemistry in a matched-22 pair cohort of 21 unifocal and 21 multifocal breast tumors. The expression of RIP140, LCoR and ERβ was inversely correlated with overall survival only for the unifocal group. The negative correlation with survival was more pronounced in patients whose unifocal tumors expressed high levels of both RIP140 and ERβ, indicating RIP140, LCoR and ERβ my play important roles in unifocal breast cancers. Some comments are illustrated below.

1: Compared to the previous study, Multicentric and multifocal versus unifocal breast cancer: differences in the expression of E-cadherin suggest differences in tumor biology. BMC Cancer 2013, 13, 361, doi:10.1186/1471-2407-13-361, the patients size was 112. The present study only obtained 42 patients. The authors need to expand the patient size.

2: I am wondering what the main proteins that regulated by RIP140 and LCoR transcriptional factors are. Will E-cadherin, N-cadherin or Vimentin involved in this unifocal breast cancer death?

3: Since the patient size are too small. It is difficult to determine that RIP140, LCoR and ERβ expression actually affect unifocal breast cancer death.

Author Response

The authors analyzed the RIP140 and LCoR levels by immunohistochemistry in a matched-22 pair cohort of 21 unifocal and 21 multifocal breast tumors. The expression of RIP140, LCoR and ERβ was inversely correlated with overall survival only for the unifocal group. The negative correlation with survival was more pronounced in patients whose unifocal tumors expressed high levels of both RIP140 and ERβ, indicating RIP140, LCoR and ERβ my play important roles in unifocal breast cancers. Some comments are illustrated below.

Comment #1: Compared to the previous study, Multicentric and multifocal versus unifocal breast cancer: differences in the expression of E-cadherin suggest differences in tumor biology. BMC Cancer 2013, 13, 361, the patients size was 112. The present study only obtained 42 patients. The authors need to expand the patient size.

Answer: The comment of Reviewer 1 allowed us to realize that this point was not clear as we used the same patient cohort as described in our former publication, with two groups framed and investigated. Indeed, based on a consecutive patient cohort consisting of 112 patients documented and surgically treated for primary breast cancer between 2000 and 2002, our total initial collective was formed by 57 unifocal breast cancer patients and 55 patients with multicentric/multifocal disease. From this collective, two equivalent groups of 23 breast cancer patients with multicentric/multifocal vs. unifocal tumors were selected using a matched paired analysis in the reference cited above. The Institutional Review Board of the Ludwig Maximilians University Munich, Germany, approved the study and all the patients gave informed consent. From this 2 groups of 23 patients, only tumors of 2 groups of 21 patients could be stained for RIP140. As this point was not clearly explained in the initial manuscript, we improved this in the “Collective” section of the Materials & Methods of the revised manuscript.

Comment #2: I am wondering what the main proteins that regulated by RIP140 and LCoR transcriptional factors are. Will E-cadherin, N-cadherin or Vimentin involved in this unifocal breast cancer death?

Answer: RIP140 and LCoR are transcription coregulators which affect the activity of a large number of DNA-binding transcription factors including nuclear receptors and others such as KLF6 for LCoR (Calderon et al. J Biol Chem. 2012 Mar 16;287:8662-74) or E2Fs, NFKB and TCF, for RIP140 (Lapierre et al. Biochim Biophys Acta. 2015 Aug;1856(1):144-50). Accordingly, the number of genes whose expression is controlled by RIP140 and/or LCoR is very large. We are in the process of characterizing these target genes in breast cancer cells by RNASEQ following knock-down of RIP140 or LCoR (S. Jalaguier and V. Cavaillès, unpublished data).

We previously reported that LCoR was correlated with NCAD expression in a large cohort of 310 breast cancers (Sixou et al. Transl Oncol. 2018 Oct;11(5):1090-1096). This marker was not analyzed in the 42 patients cohort used in the present study and it might be interesting to determine whether the correlation varies in unifocal tumors versus multifocal ones.

As already described in our publication (Weissenbacher et al, BMC Cancer 2013, reference 10 in the revised manuscript), E-cadherin, and b-catenin were correlated in the unifocal group of patients, and E-cadherin only was shown to be down-regulated in tumors in multifocal patients compared to unifocal one. Cytoplasmic b-catenin was associated with reduced OS in unifocal patients only. However, no significant correlation was observed between RIP140, LCoR or ERb expression and these 3 cell adhesion-related glycoproteins compared in the unifocal vs. multifocal groups (see Table below). These points are now mentioned in the Discussion section, and results presented as data not shown.

Table for Reviewer. Correlation analysis of RIP140, LCoR and ERb expression with cell adhesion glycoproteins expression in unifocal versus multifocal breast cancers.

Pearson coefficient

RIP140

LCoR

ERb

Unifocal

Unifocal

Unifocal

Unifocal

Unifocal

Multifocal

E-cadherin

0.179

0.082

0.082

-0.078

0.115

0.266

b-catenin

Nuclear

Cytoplasmic

0.188

0.424

-0.020

0.283

-0.020

0.283

-0.165

0.118

0.234

0.298

0.020

0.247

Muc1

Nuclear

Cytoplasmic

0.108

0.046

0.100

0.171

0.100

0.171

-0.192

0.186

0.106

0.267

-0.396

-0.165

Comment #3: Since the patient size are too small. It is difficult to determine that RIP140, LCoR and ERβ expression actually affect unifocal breast cancer death.

Answer: We are aware of the small study size, but on the other hand, we reached perfect matching criteria allowing a direct comparison of each factor in the two different groups.

Data were entered into the database in a coded fashion. Our total collective of 112 patients included 57 unifocal breast cancer patients and 55 cases of multicentric/multifocal tumors. Because of the uneven distribution of prognostic factors in our original patient group of 46 cases that met the match criteria, a matched pair analysis was performed.

A total of 23 pairs of patients, each consisting of one patient with unifocal and one with multicentric/multifocal tumor lesions, were selected according to the highest degree of equivalence in the following hierarchical and sequential order: tumor size at the time of primary diagnosis, histology grading, and lymph node status. Each parameter was required to have a p value > 0.50 to achieve intergroup homogeneity. We deliberately matched patients based on the criteria at the time of primary diagnosis. The computer software ‘Statistical Package for the Social Sciences 15.0’ (SPSS Inc., Chicago, IL, USA) was used to perform statistical analyses. We used Kruskal-Wallis one-way analysis of variance to analyze our data, which is a non-parametric method for testing equality of population medians among groups. It is an extension of the Mann–Whitney U test to 3 or more groups. From this 2 groups of 23 patients, only tumors of 2 groups of 21 patients could be stained for RIP140. These points also appear in the “Collective” section of the Materials & Methods of the revised manuscript.

To reinforce the conclusions of the study on the link between RIP140 / ERb expression and unifocal BC patient survival, we analysed the recurrence free survival. As the enhanced prognostic factor of RIP140 and ERb in the unifocal group is significant not only for the overall survival but for the recurrence free survival also, we believe this brings a new input in the results, increasing the relevance of the results. These new data are presented in the Supplementary Figure A1 (see below), as Appendix A in the revised manuscript. The results are presented and discussed in the revised “Abstract”, “Results”, “Discussion” and “Materials and Methods”.

Besides, the study is now presented with its limitations and as a preliminary report in the title, the “Abstract” and the “Discussion” of the revised manuscript.

Reviewer 2 Report

the manuscript titled “Prognostic relevance of RIP140 and ERβ expression in unifocal versus multifocal breast cancer” investigated the possible role of RIP140/ERβ as independent negative prognostic factor in breast cancer. The text clearly expresses the aim of the study and the results obtained. Despite of the small number of samples collected (42 in 2 years), the results seem convincing and the well balanced matched pair cohort is a key factor in this study. Concerning graphic abstract, it is meaningful but not significant in better and rapidly understanding the content of the article. The statistical methods section needs to be better clarified and the authors need to better illustrate the cut-off discovery procedure. The survival analysis is convincing, but the authors should also correct for other key prognostic factors (i.e. multivariable Cox Analysis) and I would suggest to expand also to an intermediate endpoint such as disease free survival. Overall the study is well designed and the results clearly presented, but the statistical section needs to be improved. The paper may be, therefore, published after major revisions

Author Response

The manuscript titled “Prognostic relevance of RIP140 and ERβ expression in unifocal versus multifocal breast cancer” investigated the possible role of RIP140/ERβ as independent negative prognostic factor in breast cancer. The text clearly expresses the aim of the study and the results obtained. Despite of the small number of samples collected (42 in 2 years), the results seem convincing and the well balanced matched pair cohort is a key factor in this study.

The paper may be, therefore, published after major revisions

Comment #1: Concerning graphic abstract, it is meaningful but not significant in better and rapidly understanding the content of the article.

Answer: We agree with the Reviewer that the graphical abstract could be improved for more clarity. Because our data are focused on the survival analysis, as preliminary report without molecular mechanism to detail, we propose either to add no graphical abstract, or to use a graphical abstract based on the 2 opposite overall survival curves of Figure 4E and 4F, as suggested in the revised manuscript. We would appreciate the advice of the editor on this point.                              

Comment #2: The statistical methods section needs to be better clarified and the authors need to better illustrate the cut-off discovery procedure. Overall the study is well designed and the results clearly presented, but the statistical section needs to be improved.

Answer: Regarding the cut-off determination, more information is given in the revised version of the manuscript (both in the Results and in the section of the Materials & Methods), explaining that the maximum difference between sensitivity and specificity, in the “survival analysis”.

Regarding the patient recruitment, because the data below were not clearly explained in the initial manuscript, we developed them in the “Collective” section of the Materials & Methods of the revised manuscript:

Data regarding the collective were entered into the database in a coded fashion. Our total collective of 112 patients included 57 unifocal breast cancer patients and 55 cases of multicentric/multifocal tumors. Because of the uneven distribution of prognostic factors in our original patient group of 46 cases that met the match criteria, a matched pair analysis was performed. A total of 23 pairs of patients, each consisting of one patient with unifocal and one with multicentric/multifocal tumor lesions, were selected according to the highest degree of equivalence in the following hierarchical and sequential order: tumor size at the time of primary diagnosis, histology grading, and lymph node status. Each parameter was required to have a p value > 0.50 to achieve intergroup homogeneity. We deliberately matched patients based on the criteria at the time of primary diagnosis. The computer software ‘Statistical Package for the Social Sciences 15.0’ (SPSS Inc., Chicago, IL, USA) was used to perform statistical analyses. We used Kruskal-Wallis one-way analysis of variance to analyze our data, which is a non-parametric method for testing equality of population medians among groups. It is an extension of the Mann–Whitney U test to 3 or more groups.

All matching criteria (tumor size, histology grade and lymph node status) were equally distributed between the two groups (p = 1.0). No significant difference was observed between the two groups in terms of age (p = 0.104 in the matched group and p = 0.533 in the total collective) or menopausal status (MG: p = 0.291 and TC: p = 0.503). Regarding histological types of tumors, the total collective (TC) demonstrated a statistically significant difference with p = 0.003, whereas no significant difference was found in the matched group (p = 0.120).

Comment #3: The survival analysis is convincing, but the authors should also correct for other key prognostic factors (i.e. multivariable Cox Analysis) and I would suggest to expand also to an intermediate endpoint such as disease free survival.

 Because of the small number of patients in each group, multivariate analysis could not be performed. Nonetheless, we agree that the analysis of the recurrence free survival might have a real input in this study. As the enhanced prognostic factor of RIP140 and ERb in the unifocal group is significant not only for the overall survival but for the recurrence free survival also, the relevance of the results is definitely increased. The DFS results are presented in the Supplementary Figure A1 (see below), as Appendix A in the revised manuscript. These new data are presented and discussed in the revised “Abstract”, “Results”, “Discussion” and “Materials and Methods”. Besides, the study is now presented with its limitations and as a preliminary report in the title, the “Abstract” and the “Discussion” of the revised manuscript.

Supplementary Figure A1.Recurrence free survival according to the expression of RIP140 and ERβ, alone or combined. Kaplan-Meier analysis of the correlation between low vs. high expression of two markers with RFS in the matched pairs (unifocal for A, C, E and multifocal for B, D, F). The analysis was performed with RIP140 (A, B) or ERβ (C, D) expression, or with RIP140 expression combined with ERβ expression (E, F). Correlations are statistically significant for p<0.01 (**) and for p<0.001 (***).

Reviewer 3 Report

In this manuscript, Mülleret al., studied the expression of two nuclear receptortranscriptional coregulators, RIP140 and LcoR in unifocal versus multifocal breast cancers by immunohistochemistry in a matchedpair cohort of 21 unifocal and 21 multifocal breast tumors. The expression ERα and ERβ was studied in parallel. The results are explained in terms of protein correlations and patient survival. The study appears to be conducted well and data explained reasonably. However, the sample size is low and immunohistochemical studies have limitations on quantitative assessment. Therefore, this manuscript has to be revised with explanation on the limitations of this study and as a preliminary report.

Author Response

In this manuscript, Müller et al., studied the expression of two nuclear receptor transcriptional coregulators, RIP140 and LcoR in unifocal versus multifocal breast cancers by immunohistochemistry in a matched pair cohort of 21 unifocal and 21 multifocal breast tumors. The expression ERα and ERβ was studied in parallel. The results are explained in terms of protein correlations and patient survival. The study appears to be conducted well and data explained reasonably.

Comment #1: However, the sample size is low and immunohistochemical studies have limitations on quantitative assessment. Therefore, this manuscript has to be revised with explanation on the limitations of this study and as a preliminary report.

Answer: We are aware of the small study size, but on the other hand, we reached perfect matching criteria allowing a direct comparison of each factor in the different groups.

Regarding the patient recruitment, because the data below were not clearly explained in the initial manuscript, we developed them in the “Collective” section of the Materials & Methods of the revised manuscript:

Data were entered into the database in a coded fashion. Our total collective of 112 patients included 57 unifocal breast cancer patients and 55 cases of multicentric/multifocal tumors. Because of the uneven distribution of prognostic factors in our original patient group of 46 cases that met the match criteria, a matched pair analysis was performed. A total of 23 pairs of patients, each consisting of one patient with unifocal and one with multicentric/multifocal tumor lesions, were selected according to the highest degree of equivalence in the following hierarchical and sequential order: tumor size at the time of primary diagnosis, histology grading, and lymph node status. Each parameter was required to have a p value > 0.50 to achieve intergroup homogeneity. We deliberately matched patients based on the criteria at the time of primary diagnosis. The computer software ‘Statistical Package for the Social Sciences 15.0’ (SPSS Inc., Chicago, IL, USA) was used to perform statistical analyses. We used Kruskal-Wallis one-way analysis of variance to analyze our data, which is a non-parametric method for testing equality of population medians among groups. It is an extension of the Mann–Whitney U test to 3 or more groups. From this 2 groups of 23 patients, only tumors of 2 groups of 21 patients could be stained for RIP140.

Besides, the study is now presented with its limitations (low number of patients and immunostainings as semi-quantitative) and as a preliminary report in the title, the “Abstract” and the “Discussion” of the revised manuscript.

Reviewer 4 Report

This article is written well, but lacks in new knowledge and grounds. Therefore, I require Major Revision.

Major point

・Please describe the analysis method of this study in detail in material and methods section.

・Figures are not unclear, replacement is necessary (Figure 1).

The reproducibility of the experiment is not clear. Please prove this.

・To prove this conclusion, data is scarce. Please prove by additional experiment (cell line, in vitro, in vivo), etc.

・And, please describe the discussion in more detail.

Minor point

・The sentence of this paper has many careful mention errors. Please review it.

Author Response

This article is written well, but lacks in new knowledge and grounds. Therefore, I require Major Revision.

Major points

Comment #1: Please describe the analysis method of this study in detail in material and methods section.

Answer: Regarding the cut-off determination, more information is given in the revised version of the manuscript (both in the Results and in the section of the Materials & Methods), explaining that the maximum difference between sensitivity and specificity, in the “survival analysis”.

Regarding the patient recruitment, because the data below were not clearly explained in the initial manuscript, we developed them in the “Collective” section of the Materials & Methods of the revised manuscript:

Data were entered into the database in a coded fashion. Our total collective of 112 patients included 57 unifocal breast cancer patients and 55 cases of multicentric/multifocal tumors. Because of the uneven distribution of prognostic factors in our original patient group of 46 cases that met the match criteria, a matched pair analysis was performed. A total of 23 pairs of patients, each consisting of one patient with unifocal and one with multicentric/multifocal tumor lesions, were selected according to the highest degree of equivalence in the following hierarchical and sequential order: tumor size at the time of primary diagnosis, histology grading, and lymph node status. Each parameter was required to have a p value > 0.50 to achieve intergroup homogeneity. We deliberately matched patients based on the criteria at the time of primary diagnosis. The computer software ‘Statistical Package for the Social Sciences 15.0’ (SPSS Inc., Chicago, IL, USA) was used to perform statistical analyses. We used Kruskal-Wallis one-way analysis of variance to analyze our data, which is a non-parametric method for testing equality of population medians among groups. It is an extension of the Mann–Whitney U test to 3 or more groups.

All matching criteria (tumor size, histology grade and lymph node status) were equally distributed between the two groups (p = 1.0). No significant difference was observed between the two groups in terms of age (p = 0.104 in the matched group and p = 0.533 in the total collective) or menopausal status (MG: p = 0.291 and TC: p = 0.503). Regarding histological types of tumors, the total collective (TC) demonstrated a statistically significant difference with p = 0.003, whereas no significant difference was found in the matched group (p = 0.120). 

From this 2 groups of 23 patients, only tumors of 2 groups of 21 patients could be stained for RIP140.

Comment #2: Figures are not unclear, replacement is necessary (Figure 1).

Answer: We agree with Reviewer 3 as the photographs had been pasted as.pdf in the initial manuscript with a low resolution. In the revised Figure 1, the photographs appear with the correct resolution and are not blurry.

Comment #3: The reproducibility of the experiment is not clear. Please prove this.

Answer: We did not mention in the initial manuscript that for several tumors (14 out of 42), RIP140 and LCoR stainings were performed 3 times by 3 independent persons to check the reproducibility of the stainings and of the analysis. This point is now added in the Immunohistochemistry section with the fact that for all sections, the stainings were analyzed by two independent observers.

We also prepared for the Reviewer a Figure (see below) with the negative and positive controls of RIP140 and LCoR stainings.

Besides, the small number of patients in each group is now explained in the revised manuscript (see answer to comment #1 above). Moreover, we state clearly in the last paragraph of the revised “Discussion” that the results should be investigated in extended cohorts, as no interindividual reproducibility is possible with a patient collective.

Finally, the study is now presented with its limitations and as a preliminary report in the title, the “Abstract” and the “Discussion” of the revised manuscript.

Comment #4: To prove this conclusion, data is scarce. Please prove by additional experiment (cell line, in vitro, in vivo), etc

 Answer: This study is based on histoimmunostaining of tumor tissues which is a recognized technique to draw conclusions about biomarker correlation with patient survival, as long as correlations are statistically significant.

We agree that further experiments are needed to fully decipher the involvement of both RIP140 and ERb in differentially driving breast cancer progression in unifocal vs. multifocal breast cancers. However, no in vitro or in vivo biological models of unifocal and multifocal breast cancers are easily available. Moreover, no additional experimental data can be reasonably performed during the 8 days given by the editor to revise the manuscript.

But to reinforce the conclusions of the study on the link between RIP140 / ERb expression and unifocal BC patient survival, we analysed the recurrence free survival in this study. As the enhanced prognostic factor of RIP140 and ERb in the unifocal group is significant not only for the overall survival but for the recurrence free survival also, we believe this brings a new input in the results, increasing the relevance of the results. These new data are presented in the Supplementary Figure A1 (see below), as Appendix A in the revised manuscript. The results are presented and discussed in the revised “Abstract”, “Results”, “Discussion” and “Materials and Methods”.

Comment #5: Please describe the discussion in more detail.

Answer: We find difficult to discuss our results in more details as we went again through the literature  and found no relevant mechanistic hypothesis which would oppose unifocal versus multifocal cancer cell signaling pathways. We prefer to keep it focused on the link between RIP140 and ERb mechanisms of action as it appeared in the last paragraph of the Discussion. However, to take into account the referee’s comment, we have strengthen the discussion section by adding at the end, the section “5.Conclusion” which was only mandatory.

Moreover, as already described in our publication (Weissenbacher et al, BMC Cancer 2013, reference 10 in the revised manuscript), E-cadherin, and b-catenin were correlated in the unifocal group of patients, and E-cadherin only was shown to be down-regulated in tumors of multifocal patients compared to unifocal one. Cytoplasmic b-catenin was associated with reduced OS in unifocal patients only. Nonetheless, no significant correlation was observed between RIP140, LCoR or ERb  expression and these 3 cell adhesion-related glycoproteins compared in the unifocal vs. multifocal groups (see Table below). This point is mentioned and more detailed in the Discussion section as data not shown.                          

We hope that the referee will now find the Discussion long and interesting enough.

Minor point

Comment #6: The sentence of this paper has many careful mention errors. Please review it.

Answer: We understand for this comment that the references (and not “sentence”) have to be checked. We agree that 2 references were not useful in the previous manuscript, with a shift of the numbering of the following references. We carefully checked all the references that are now adequate in the revised manuscript.                               

Supplementary Figure A1. Recurrence free survival according to the expression of RIP140 and ERβ, alone or combined. Kaplan-Meier analysis of the correlation between low vs. high expression of two markers with RFS in the matched pairs (unifocal for A, C, E and multifocal for B, D, F). The analysis was performed with RIP140 (A, B) or ERβ (C, D) expression, or with RIP140 expression combined with ERβ expression (E, F). Correlations are statistically significant for p<0.01 (**) and for p<0.001 (***).

Round  2

Reviewer 1 Report

All questions has been answered.

Reviewer 2 Report

The authors have addressed all reported issues, they moreover successfully highlighted all limitations of their study.

The current graphical abstracts' solution is convincing.

In my opinion, the manuscript is now suitable for publication.

Reviewer 4 Report

This manuscript has been improved by amendment.